# Climate Change, Extreme Temperatures and Sex-Related Responses in Spiders

**DOI:** 10.3390/biology12040615

**Published:** 2023-04-18

**Authors:** Jeffrey A. Harvey, Yuting Dong

**Affiliations:** 1Department of Terrestrial Ecology, Netherlands Institute of Ecology, Droevendaalsesteeg 10, 6708 PB Wageningen, The Netherlands; 2Department of Ecological Sciences, Section Animal Ecology, Vrije Universiteit Amsterdam, De Boelelaan 1085, 1081 HV Amsterdam, The Netherlands

**Keywords:** adaptation, climate change, ectotherm, fitness, heat, survival

## Abstract

**Simple Summary:**

Anthropogenic climate change is one of the greatest threats to biodiversity. Extreme temperature events associated with longer-term climate change are increasing in frequency, duration and intensity. The effects of climatic extremes on ectotherms, such as insects, have been well-studied in recent years. However, the effects of extreme temperatures on other arthropod groups, such as spiders, has received much less attention. Spiders are important organisms as predators in natural and agricultural ecosystems. In this paper, we describe spider responses to extreme temperatures and highlight the most important knowledge gaps that urgently need to be filled to better understand how vulnerable spiders are to climate change and climatic extremes. Unlike insects, traits such as body size and niche breadth may differ markedly in male and female spiders. Therefore, we argue that research needs to address the effects of heat exposure on the physiology, behavior and ecology of male and female spiders across multiple taxa. Observed declines in some terrestrial insects have been widely reported in recent years, with climate change, along with other anthropogenic threats, being implicated. Longer-term data on trends in spider abundance, where available, may also shed possible light on the role of climate change.

**Abstract:**

Climatic extremes, such as heat waves, are increasing in frequency, intensity and duration under anthropogenic climate change. These extreme events pose a great threat to many organisms, and especially ectotherms, which are susceptible to high temperatures. In nature, many ectotherms, such as insects, may seek cooler microclimates and ’ride out´ extreme temperatures, especially when these are transient and unpredictable. However, some ectotherms, such as web-building spiders, may be more prone to heat-related mortality than more motile organisms. Adult females in many spider families are sedentary and build webs in micro-habitats where they spend their entire lives. Under extreme heat, they may be limited in their ability to move vertically or horizontally to find cooler microhabitats. Males, on the other hand, are often nomadic, have broader spatial distributions, and thus might be better able to escape exposure to heat. However, life-history traits in spiders such as the relative body size of males and females and spatial ecology also vary across different taxonomic groups based on their phylogeny. This may make different species or families more or less susceptible to heat waves and exposure to very high temperatures. Selection to extreme temperatures may drive adaptive responses in female physiology, morphology or web site selection in species that build small or exposed webs. Male spiders may be better able to avoid heat-related stress than females by seeking refuge under objects such as bark or rocks with cooler microclimates. Here, we discuss these aspects in detail and propose research focusing on male and female spider behavior and reproduction across different taxa exposed to temperature extremes.

## 1. Introduction

There is increasing empirical evidence that the biosphere is in the early stages of a major extinction event that is primarily anthropogenic [1,2,3,4]. Around 20,000 closely monitored species (vascular plants, vertebrates, invertebrates) have lost as much as 60% of genetic diversity over the past five decades alone [5]. Recent studies are also reporting that many terrestrial and aquatic insect taxa or biomass have declined rapidly over at least several decades [1,6,7,8,9]. These declines in insect numbers will impact other species in food chains, including insectivores [10]. For example, many species of insectivorous birds have also declined markedly over the past several decades, especially in temperate biomes [11,12]. The loss of insects will invariably impact entire food webs. Simplified food webs in turn will eventually hamper the functioning of ecosystems [13,14,15]. Insects also provide an array of important ecological services, such as pollination, pest control and nutrient cycling [16,17]. The problem of insect declines is therefore acknowledged as a major threat to ecosystems and human well-being [18].

An array of human-mediated factors is assumed to be driving biodiversity loss, including habitat destruction and fragmentation, invasive species, chemical and other forms of pollution, overharvesting and anthropogenic climate change (ACC). The ecological effects of ACC on biodiversity is receiving increasing attention in the scientific literature [19,20,21,22]. Rapid shifts in climate-related parameters such as temperature and rainfall have allegedly played a major role in several previous mass extinction episodes [23,24]. Despite this, the relative importance of ACC in driving biodiversity loss is open to conjecture, although there is little doubt that it may act in synergy with other anthropogenic stresses [25]. There is a general consensus that if warming continues unabated, it will exacerbate biodiversity loss and the extinction crisis [26].

Ectothermic organisms, including insects and other arthropods such as spiders, are especially susceptible to rapid temporal changes in temperature due to their small size and because body temperature is closely linked with ambient temperature [27,28,29]. Temperature is important in that it can affect the metabolic expenditure of resources, and if conditions become too hot or cold, this can reduce fitness through a decrease in fecundity or via precocious mortality [30,31]. Some arthropod species, such as those living in deserts or tundra, possess physiological adaptations to endure diel or seasonal exposure to extreme heat or cold [32,33,34], whereas species which live in more stable or predictable climates do not [35]. Many arthropod species have life cycles that vary seasonally, such as annual species that overwinter in diapause as eggs or pupae and larvae or adult stages that are active in spring and summer. Many Palearctic and Nearctic insects disperse northwards in spring but migrate southwards in late summer or autumn to warmer regions where they overwinter as adults in diapause [36]. Thermophilic insects that remain in northerly parts of their summer range are often killed by deep winter frosts [37].

Evidence is accumulating that insects are responding to gradual (long-term) warming through range and elevational shifts, changes in seasonal and/or diel growth and activity patterns and related processes such as seasonal voltinism [38,39,40,41,42,43]. Embedded in gradual longer-term warming, however, are climatic extremes (CE), such as heatwaves, droughts and heavy precipitation events, as well as attendant events such as floods and fire, that generally occur over short timescales but have been increasing in frequency, duration and intensity over the past two decades [44,45,46,47,48,49]. CE are gaining increased attention for their effects on biodiversity at all levels of organization [50,51,52]. Of the various types of CE, heat waves, in particular, may be lethal to insects if critical temperature or temporal exposure thresholds are exceeded [53,54]. More recent analyses reveal that critical high temperatures for insect survival (the ‘critical thermal limit’, CTL) are higher than for reproduction (the ‘thermal fertility limit’, TFL) in insects such as fruit flies and flour beetles [55,56]. Male insects seem particularly susceptible to heat, which apparently leads to death of sperm and precocious sterility [57,58,59]. Increased duration to heat exposure can also affect operational sex ratios in fruit flies [60]. A recent meta-analysis [61] shows that insects in general respond negatively to CE. A major concern is therefore that CE are pushing many insect species beyond their adaptive limits [62], exposing them to conditions that they may not have experienced in their evolutionary history. Despite this, vast knowledge gaps remain in our understanding of the eco-physiological effects of CE on insects, and effects on other terrestrial arthropod groups, such as spiders, have been little studied.

## 2. Spiders and Climate Change

Spiders (Araneae) are abundant and ubiquitous predatory arthropods that occur in most terrestrial (and even some freshwater) ecosystems, and by consuming huge amounts of insect biomass, they play an important role in ecosystem functioning and biological control [63,64]. Despite this, little is known about even the basic aspects of the biology and ecology of most spider species. This is particularly true in terms of demographic aspects, where there are few if any data on temporal trends in the abundance and/or biomass of spiders in different regions or habitats in response to abiotic factors linked to anthropogenic stresses such as habitat loss, pesticides or ACC. The effects of ACC and CE on spiders are therefore restricted to a few studies on the physiology or behavior of individual species or genera in response to heat exposure, with less data available on longer-term abundance or distributional shifts. A few studies are reporting that warming is correlated with potential or realized shifts in spider distribution northwards [65,66] or with a decrease in niche space or habitat suitability within the current range [67,68]. In thermally stable habitats, such as in caves, some authors are suggesting that rapid warming may drive many spiders adapted to these habitats to extinction in the coming decades [69]. Clearly, much more data are required in order to determine if alarming terms such as ‘insectageddon’ or ‘insect apocalypse’, that are sometimes used to describe insect declines across the biosphere [70,71,72], also apply to spiders.

### 2.1. Sexual Size-Dimorphism and Ecological Variation in Male and Female Spiders

Spiders are amongst the most unique of arthropods in that there is considerable phylogenetic variation in the expression of sex-related life-history traits such as body size [73]. For instance, the greatest variation in sexual size dimorphism in the animal kingdom is found in spiders [74,75]. The body mass ratio of adult males to females in some lineages is 50:1 or even greater [73,75]. Extreme sexual size dimorphism (SSD), characterized by female gigantism and/or male dwarfism, appears to be prevalent in several spider families, such as the Araneidae, Tetragnathidae and Theridiidae [76,77,78,79] (Figure 1A,B). However, even within those families, there is considerable variation in the degree of SSD [80], and there is even greater variation in SSD among spiders in different families. For instance, male spiders in families such as the Lycosidae, Zoropidae, Pisauridae and Salticidae are often only fractionally smaller than females (Figure 1C,D). Most females of spiders in these families do not construct webs, but both sexes are sit-and-wait ambush predators, or hunt for prey cursorially, co-occurring in the same habitats on the ground or in vegetation [81,82,83].

Several factors have been posited to explain the extreme variation of SSD exhibited by spiders in different families, and these factors may be strongly linked with phylogeny. For instance, when females make webs high in vegetation, reduced mass-specific power and gravity are greater impediments in large rather than in small males that must climb to locate mates [84,85,86,87,88]. On the other hand, male and female spiders in many families often differ significantly in their ecology, with females constructing webs and having a sedentary lifestyle while males may move considerable distances in search of mates [89,90,91]. The risk of mortality from visually foraging predators is presumably higher in wandering males than in sedentary females sitting effectively motionless in their webs; hence, selection favors males attaining precocious sexual maturity at an earlier (and smaller) stage in their development than females, which may benefit from larger size by achieving higher fecundity [74,92,93]. On the other hand, when competition amongst males for access to mates is high, and/or when both sexes are under similar risks to survival such as predation, then selection may favor an increase in male size and thus reduced SSD [92]. In some wolf spiders, females search for males living in burrows and initiate courtship, and SSD in these species is reversed, with males being the larger sex [94].

The importance of climate-related factors, such as extreme temperatures, in relation to the costs and benefits of SSD between male and female spiders has so far not been addressed. SSD is not only characterized by differences in body size but in the expression of other morphological traits. Where SSD is extreme, females often have bulky bodies and short appendages, which are traits that are adapted to low motility and high fecundity. Males, on the other hand, often possess longer appendages (relative to body size) than females and more streamlined bodies, making them less cumbersome and more motile [95]. Therefore, sites selected by adult females in which to construct prey-capturing webs will invariably be strongly affected by microclimatic factors (see below), whereas adult males will possess more plastic responses in this regard. Once the web is constructed, the female adopts a largely sedentary lifestyle, and her vertical and horizontal movement is limited. Males, on the other hand, can seek out various types of microhabitats during CE, such as heat and intense rainfall. This may make males less prone to the negative effects of CE than females. It is important to stress that variation in body size is often much more variable in male than in female spiders, given that the optimal phenotype in male spiders may be based on trade-offs between the costs and benefits of size and development time. Although variation in body size is sometimes multi-fold in some species [91], how variation in SSD—from extreme to low—affects susceptibility to abiotic factors such as temperature is largely unknown (see below).

### 2.2. Physiological and Behavioral Responses of Spiders to High Temperatures

As with insects, spider metabolism is strongly affected by abiotic conditions such as temperature and moisture [96], and this often correlates with habitats in which the spiders are found [97]. In hot years, higher temperatures at the warmest urban locations were found to negatively affect the abundance of ghost spiders (Anyphaneidae), facilitating an increase in more thermally tolerant insect tree foliage-feeding herbivores in Raleigh, North Carolina, United States [98]. Spiders inhabiting dry, arid habitats, such as deserts, exhibit physiological adaptations to these conditions that are sometimes lacking in temperate relatives [99]. Unsurprisingly, exposure to heat and other abiotic stresses such as drought can increase respiration rates in spiders while decreasing survival or altering development [100]. The wolf spider *Pardosa glacialis* is found in Arctic ecosystems, and each sex responds differently to inter-annual variations in temperature, with larger adult size and the degree of SSD increasingly skewed toward larger females when snowmelt occurs earlier in spring [101]. High temperatures can also generate a range of behavioral shifts or affect development and survival. For example, exposure to heat resulted in high mortality of the American house spider, *Parasteatoda tepidariorum*, and both higher juvenile mortality and extended development time in the Western black widow, *Latrodectus hesperus* [102,103,104]. All eggs of two invasive widow spiders (*Latrodectus* spp.) in Japan that were exposed to extreme heat for only 10 min failed to hatch [105]. Brown recluse spiders, *Loxosceles reclusa*, were unable to survive for more than 130 min if exposed to temperatures of 48 °C [106]. In the social spider, *Anelosimus studiosus*, exposure to warmer temperatures affected individual behavior by elevating activity, reducing tolerance toward conspecifics and latency in attacking prey, thus increasing a tendency to attack other spiders [107]. *Latrodectus hesperus* spiderlings that were exposed to high temperatures were also more likely to engage in the cannibalism of siblings [108]. Different phases of courtship behavior in the desert jumping spider, *Habronattus clypeatus* also varied in duration when male and female spiders were kept at room (25 °C) or high (50 °C) temperatures, and this did affect copulatory success [109].

The above studies show that the CTL for spider eggs, juveniles and adults occurs at temperatures that would represent extreme heat (i.e., over 45 °C). However, spiders may also respond to high temperatures by exhibiting changes in other traits such as behavior and development. A recent meta-analysis revealed that the average body size of spider assemblages increased from cool/moist to warm/dry environments, and it was accounted for by a turnover in body size from small-bodied to large-bodied spider families [110]. However, the relationship between body size and climate was inconsistent within families [110]. One of the questions arising from this study is how SSD is accounted for, taking phylogeny into consideration, and it assumes that many studies simply focus on female body size. Sexual dimorphism in spiders has generally focused on differences in male and female body size and how this is correlated with biotic and abiotic selection pressures that differentially affect each sex. This is especially true in web-building spiders, where females have well-defined silk glands, using them to construct prey-capturing webs and to wrap newly ensnared prey in them, or in the production of egg sacs. These females spend most of their lives in or close to their webs, and thus, they exhibit a limited capacity to move beyond a restricted micro-habitat. On the other hand, male spiders live in prey-capture webs only during juvenile development and abandon them as adults, adopting a cursorial, nomadic lifestyle in which they search for mates [111].

As discussed earlier, the spatial area of habitat inhabited by web-building and effectively sessile females is generally considerably smaller than in wandering males. The selection of habitats in which females construct their webs is critical in determining how exposed they are to sunlight and heat stress. Over evolutionary time, adult females of orb-web spiders have expanded their habitats from shady, cool locations to more open, brighter, warmer environments [112]. The sedentary lifestyle of female spiders exposed to sunlight has generated strong selection pressures to deal with thermal stress. This may include a shift to lighter colors on the cephalothorax and abdomen, such as white, yellow and silver that more effectively reflect heat (Figure 2), or changes in the surface area to volume ratio (SVR). However, the spectral reflectance efficiency and SVR of 11 spider species in four genera did not differ significantly between congeners inhabiting bright or shady environments, even though the former group had higher lethal temperatures [113]. This suggests that spiders living in brighter, sun-exposed habitats exhibit different physiological or behavioral adaptations to high temperatures. Some spider species that make their webs in shady or sunnier habitats alter their behavior depending on location [114]. For instance, in open habitats, the funnel-web spider *Agelena limbata* reduces foraging activity when webs are exposed to direct sunlight and temperatures exceed 40 degrees, whereas spiders living in cooler, wooded habitats do not [115]. Food consumption of adult spiders in the open habitat was significantly lower than in the woody habitat. Extreme heat or extended heat waves may also confer costs to male spiders, which are forced to seek shelter and remain quiescent under these conditions, whereas when it is cooler, they are better able to search for mates and food. The surface area to volume ratio in male spiders, with small bodies, is higher than in females, with larger, compact bodies, also making them more susceptible to heat. Furthermore, model simulations also suggest that extreme heat or humidity have negative effects on the quality of silk in the webs of orb-web spiders (*Argiope* spp.), directly or indirectly reducing web capture performance [116]. Under these conditions, females that make exposed webs (i.e., elevated orb webs that are connected to vegetation) may be under much greater selection to either adapt to or avoid exposure to direct sunlight than spiders, which habitually live on the ground or make smaller webs under objects such as bark or rocks (Figure 3).

One important caveat is that the effects of extreme heat on spider survival and fecundity may differ with sex. In insects, it has been reported that males are often much more sensitive to heat than females, at least in terms of reproduction [55,57,58]. At given high temperatures, sperm is thus more likely to be killed in the testes of males than eggs are in the ovaries of females. Thus far, however, the effects of heat on sperm in male spiders has not been studied. Mating and sperm transfer in male spiders is quite unique among arthropods. Insect reproduction involves direct insertion of the male genitalia into the female vagina and sperm are stored in specialized structures, the spermathecae [117,118]. In spiders, sperm are transferred via the genitalia into a specialized web spun by the males, and sperm in turn are taken up into specialized structures (emboli) on the terminal end of the male palps [73]. During mating, males insert their palps individually to the female genitalia (epigynum), and their sperm is transferred to the female in a coiled and encapsulated state to be stored in spermathecae [119]. De-encapsulation only occurs just before the egg is fertilized, and therefore, encapsulation of the sperm may provide some protection from heat exposure. The duration of copulation in spiders is also often prolonged, lasting many minutes [120,121], despite the fact that a full insemination can be completed much faster [121,122]. Although the duration and mechanics of copulation duration have evolved under sexual selection, amongst male and female spiders, they can be disrupted by both biotic and abiotic factors. These include the presence of multiple males competing for an individual female (where one male disrupts copulation between the female and another male), the presence of predators, sudden wind or heavy rain. The adaptive benefits of extended copulation in spiders has been discussed [121,123], but the costs have received less attention. The physiological condition of many insects is negatively affected by heat [52,53,54], and thus, it will be interesting to determine if reproduction in spiders under extended copulation is also affected under variable temperatures.

## 3. Conclusions and Future Directions

A suite of human-mediated environmental stresses is negatively affecting and altering ecosystems and the species in them across most of the biosphere. Growing evidence suggests that genetic diversity among well-studied plants and animals has eroded significantly over the past five decades [1] and that anthropogenic climate change (ACC) and climatic extremes (CE) are potentially important drivers, either acting independently or synergistically with other stresses [124]. Both act over different timescales and require different kinds of eco-physiological responses in order to adapt to them. Ectothermic organisms are especially vulnerable to CE, especially short-term heat waves. Although a growing body of literature is examining the effects of ACC and CE on insects, little is thus far known about the responses of spiders to them. Moreover, unlike with insects, long-term data on trends in the abundance of spiders across different taxa in response to global changes is scarce. Given their affinity with insects in terms of size and general biology, and the fact that as arthropods they co-occur in most terrestrial (and even aquatic) ecosystems, it is not a stretch to argue that spiders are also declining in many regions.

Many gaps in our understanding of the effects of ACC and CE in particular on spiders thus remain to be filled. As was suggested in recent commentaries, microclimates drive the vulnerability of arthropods to CE [19,125,126,127]. In highly simplified landscapes, such as in many urban parks and gardens, golf courses and agricultural fields, herbaceous vegetation has often been removed and replaced by pavement, grass that is mown regularly or crop monocultures. These habitats are effectively ‘biological deserts’, warm rapidly and have little buffering effect against extreme heat [128]. Differences in temperature between dense stands of natural vegetation and the edge of vegetation plots are also amplified during heat waves [129]. Consequently, ACC and CE need to be factored into management strategies aimed at the conservation of arthropods and revitalization of ecosystems across landscapes [19]. Future research needs to focus on the variable effects of ACC and CE, as well as other human-mediated stresses, on the biology and ecology of spiders. Given that the males and females of many spider species differ profoundly in terms of these parameters, each sex may thus be under different selection pressures in responding to short-term CE such as heatwaves and how these interact with other anthropogenic drivers of biodiversity loss.

We suggest that future research on spiders in response to ACC and other anthropogenic threats pays particular attention to the following areas: first, data sets on spider abundance and/or biomass over many years from multiple locations need to be extracted and collated so that demographic trends can be elucidated. This has been addressed with insects [6,7] in recent years, but despite their ecological importance, spiders have been generally overlooked. If reliable data are only available for specific taxa or taxonomic groups, then these should be examined. There is little reason to believe that spiders are faring any better than insects under rapid anthropogenic global changes. It is likely that multiple factors may account for changes in spider abundance, but the first step is to show that general declines in biodiversity also include changes in the abundance of spiders. Pitfall trapping has yielded immense amounts of data for ground-dwelling predatory beetles such as the Carabidae, and since many spiders (i.e., Lycosidae) also actively forage on the ground, invariably, there must be data sets available on them as well. This step needs therefore to be initiated as soon as possible. Second, CTL studies have been performed on very few spider species, and, as far as we know, nothing is known about TFL in spiders. It is therefore important to determine the physiological and reproductive effects of exposure to extreme heat in many more ectothermic arthropods, including spiders. Furthermore, it will be important to determine if the sperm of male spiders is differentially susceptible to heat-related stress during the various stages before, during and after transfer to the female. Moreover, do extreme temperatures affect the hatching success of spider eggs or the subsequent development of the spiderlings and adults? This was recently shown in a tropical butterfly [130]. Finally, studies are needed in which the behavior of both sexes of spiders that build prey-catching webs or that do not are examined under actual or simulated heat wave conditions. It will be interesting to determine if wandering males are more labile in their usage of micro-habitats than web-building females or if females deliberately choose cooler microclimates in which to construct webs.

The importance of spiders in controlling insect and other arthropod pests in cropping systems is well documented [131]. Thus, it remains an enigma why so many critical knowledge gaps exist in our understanding of the biology and ecology of spiders and how anthropogenic stresses are affecting them. These gaps urgently need to be filled in the coming years. Although our paper focuses on the effects of ACC and especially CE on spiders, it is important to examine how spiders are responding to other anthropogenic changes across the biosphere. This information will prove to be crucial in implementing strategies aimed at conserving these fascinating organisms.

## Figures and Tables

**Figure 1 biology-12-00615-f001:**
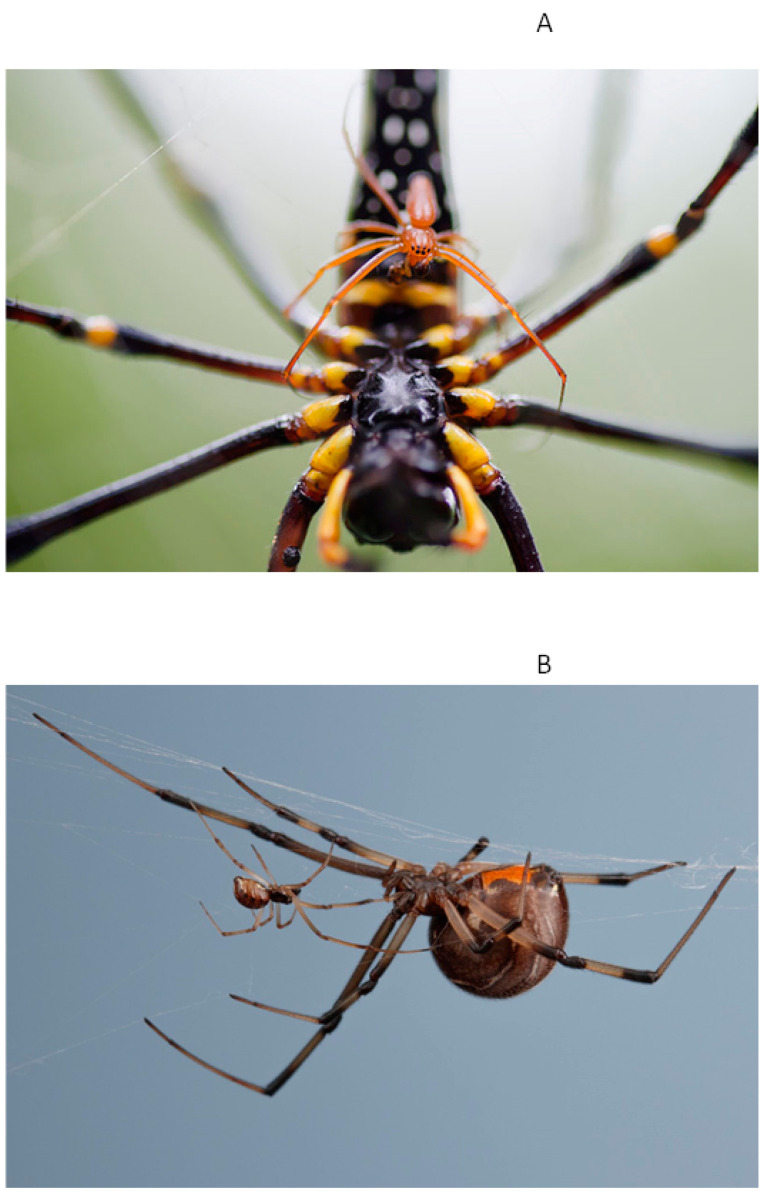
Photographs showing immense variability in the degree of sexual size dimorphism in spiders. (**A**) Giant golden orb-weaver, *Nephila pilipes* (Araneidae), adult male (orange) about to mate with a much larger adult female. (**B**) Adult male (left) courting a much larger adult female (right) of the brown widow spider, *Latrodectus geometricus* (Theridiidae). (**C**) Adult male (upper right) and adult female (lower left) of the spotted wolf spider, *Pardosa amentata* (Lycosidae), illustrating similarity in body size. (**D**) Adult male (under leaf, left) and adult female (upper leaf, right) of the jumping spider *Jotus remus* (Salticidae) The male is similarly sized with the female and uses a fan-like structure on the tarsi of the third right leg to signal females of its presence.

**Figure 2 biology-12-00615-f002:**
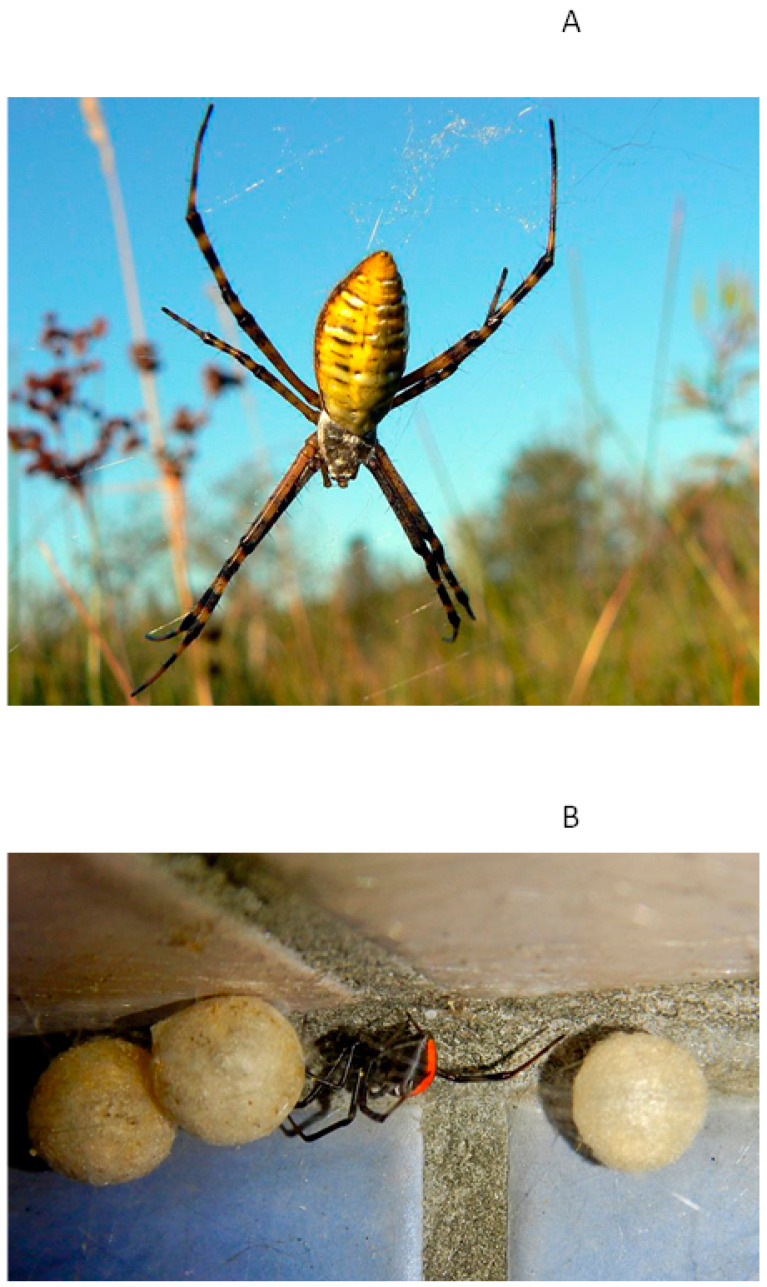
Photographs showing (**A**) Adult female of the banded argiope, *Argiope trafasciata* (Araneidae), sitting in its web and exposed in sunlight. This species is common is warm, dry climates and has body coloration (silver cephalothorax, yellow abdomen), which may help to reflect light and reduce heat absorption during the day. (**B**) Adult female of the Redback false widow spider, *Latrodectus hasselti* (Theridiidae), attending three egg sacs. This species is found warm dry climates in its native Australia but avoids bright habitats and most commonly spins small, tangled webs under rocks, fallen trees and in other dark habitats (including around human habitation). Its primarily black coloration makes it poorly adapted to open habitats and direct exposure to sunlight.

**Figure 3 biology-12-00615-f003:**
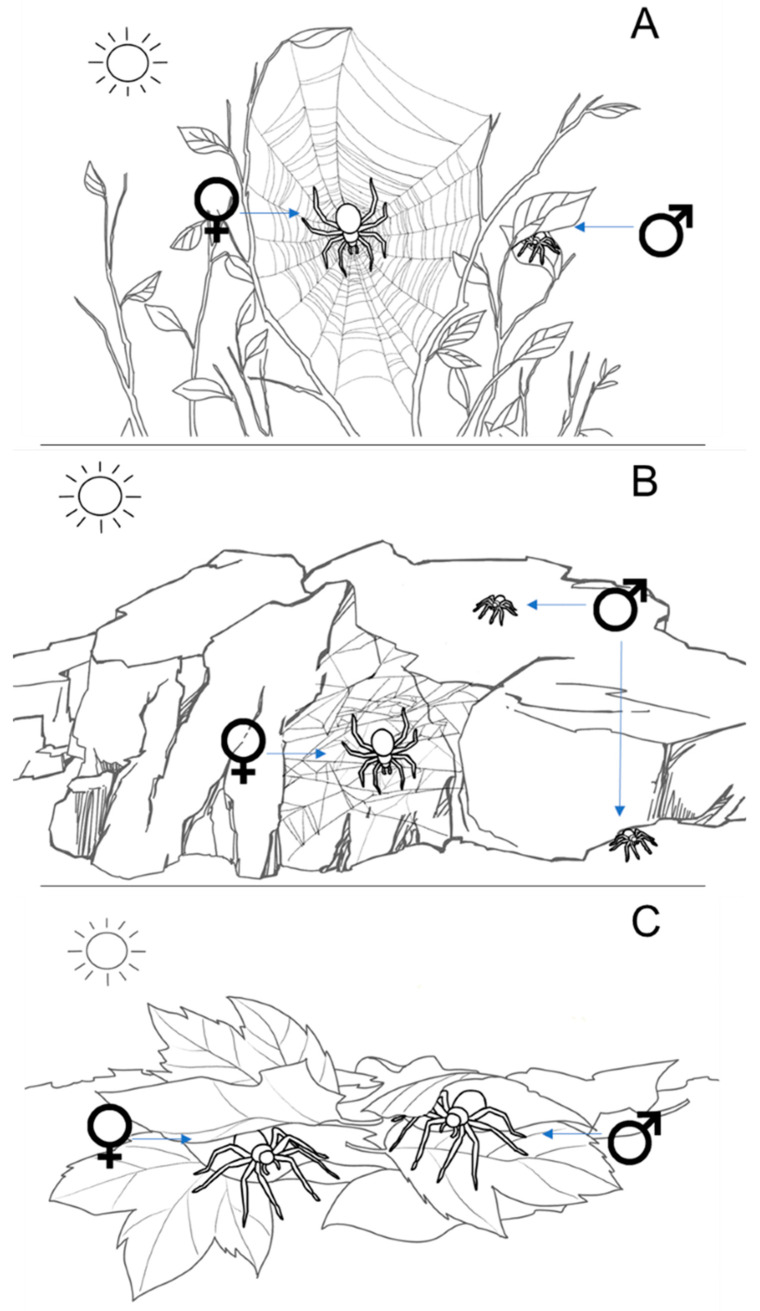
Illustration showing the differing biology and behavioral strategies of spiders in response to habitats with variable exposure to direct sunlight. Many species of web-building spiders (**A**,**B**) show significant female-biased sexual size dimorphism (SSD), and males and females may also exhibit discernible differences in behavior and ecology. In (**A**), orb-web-building adult female spiders (Araneidae) construct their large webs in open habitats and are thus often exposed to hot and bright conditions. Adult males, on the other hand, do not construct webs and instead spend most of their time searching for mates. This means they have much larger niches, which gives them the opportunity to seek shelter under rocks, bark or the canopy of plants during the day or when abiotic conditions are unfavorable. In (**B**), tangle-web weaving adult female spiders (Theridiidae) construct small webs under rocks or in crevices which are perpetually shaded. Males also frequent these habitats but may need to temporarily move into more exposed locations to search for mates. Alternatively, some non-web building spiders do not show significant SSD, and both sexes exhibit similar biology and ecology. In (**C**), male and female wolf spiders (Lycosidae) co-occur in the same microhabitats, and both sexes forage and search for mates when abiotic conditions are optimal.

## Data Availability

No new data were created in this article.

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
