# Peer review of "Climate Change, Extreme Temperatures and Sex-Related Responses in Spiders"

_biology, 2023, doi:10.3390/biology12040615_

Round 1

Reviewer 1 Report

This is a timely and useful review. I have only minor comments.

Author Response

We appreciate the referee's corrections and suggestions and these have been fully implemented into the revision (mostly in yellow font, but italics added to references). 

Reviewer 2 Report

This is a well written and organized assertion of the appropriateness of research into how spiders deal with CE and whether that differs predictable among varying life histories and sexes.  Indeed, the body of the "review" returns to these themes repeatedly, but the ideas are never developed with data examples the way they would be in a traditional review, presumably because the work still needs to be done.  As such, my primary complaint of the paper is that it is not a review, and it is not new data, so what is it and what function does it serve?  I suppose it is a call to more research in this area.  And I will leave it up to the editors whether that is what they want to publish.  From my vantage point the work is most definable as a development of one's thinking, as a prelude to a grant proposal perhaps.  I could totally see this being the justification for proposed work that tests, for example, whether cursorial females suffer less from CE than web building females, and whether males suffer less than females.  But there is not enough reviewed here, and there is not enough of a new idea advanced here for me to get too excited about the work.    

Author Response

The referee makes some very sound points. We wish to stress (as we did in the response to the editor of this special issue) that the paper is BOTH a review of existing data on spider responses to extreme temperatures, using climate change-mediated increases in extreme temperature events as our baseline, AND a perspectives paper, which emphasizes how many knowledge gaps remain before we can more fully elucidate the consequences. As we show, spiders do exhibit physiological and behavioral responses to heat exposure, but existing studies have not placed these findings in the context of climate change. Moreover, little is known about male responses to extreme temperatures, which is odd given that (as we explain) the ecology of male spiders often differs markedly from the ecology of female spiders, which sets these fascinating organisms apart from many insects, where males and females share niche space. Also, given that sexual size dimorphism is highly variable between and within spider families, it will be important to evaluate in future studies how this may affect responses to extreme temperatures in a warming world. Our new paragraph in the conclusions and future directions section explains three major areas that need to be studied in more detail, so we envisage that the paper will stimulate spider ecologists to begin addressing them. 

Reviewer 3 Report

Please find my comments in the attached file. Overall this is a nice work. However I am not sure about the title, for it seems rather misleading. I think the focus of this work is to identify the gaps in knowledge and help directing the relevant research under the immediate need to look for the effect of climate change on organisms, spiders in this case. So I would encourgae the authors to modify their title

Author Response

We thank the reviewer for his/her supportive comments. We agree that the original title is somewhat ambiguous so we deleted it and replaced it with a new one that we think is punchier and addresses more strongly the topic being discussed (yellow font). 

Reviewer 4 Report

The manuscript is announced as a review, but it may perhaps better be characterized as an essay or a commentary. The main idea is that since males and females within some spider families (but not others) have different lifestyles with respect to microhabitat and activity, they will be differentially affected by events of climatic extremes. This is a viable hypothesis, which is argued for by reference to laboratory and field studies. None of these directly address the question raised by the hypotheses, however. Therefore, the conclusion postulated in title is hardly drawn in the paper itself, which is appropriately formulated with many “may be”, “could be”…. The title should also leave it as a hypothesis, for example: “Susceptibility to climatic extremes may vary with sex and phylogeny in spiders”. The paper points on several interesting questions that are still unstudied in spiders, e.g. the susceptibility of sperm to high temperatures, both when within the males and within the female.

Fig. 3: I am not against schematic drawings made to support a theoretical point. But they should either be completely schematic or approach something naturalistic. In fig. A and B, the same drawing of a spider is smashed upon drawings of an orb-web and (what is supposed to be) a tangle-web, respectively. Luckily, the same drawing is not used to illustrate the hunting spiders in fig. C. Orb-web spiders sit head-down in the hub of their web, and tangle-web spiders hang upside-down in their tangle. I get frustrated when popular magazines show orb-web spiders sitting head-up (they sometimes even turn photos around to achieve this), but I get even more frustrated when it happens in a scientific journal by authors that are supposed to know about spiders. Either make the drawings “correct” (even if somewhat schematic) if they are supposed to illustrate specific types of animals, or delete Fig. 3 (the point is clear enough from the text). Overall, none of the figures are essential for the paper.

Minor comments

Some jumps in font size: Line 7: “De”. Line 242-9: whole section. Line 566-573: whole section. Species names not in italics.

The writing suffers from unnecessary exaggerations that are really not needed: Line 32: “exclusively anthropogenic”. It may be difficult to distinguish anthropogenic from background extinction, but that doesn’t mean that the latter is eliminated. Line 64: “invariably”. This implies that no exceptions are likely to exist.

Line 77: pleonasm, delete “,in particular,”

Line 235: are emboli “glands”? I would prefer just “structures”.

Line 239: sperm remains encapsulated in the female spermathecae; de-encapsulation does not happen until just before the eggs are fertilized.

Line 582: something is missing in “… and each sex expresses may also exhibit differences …”

Line 587: crevices?

Author Response

Reviewer four, like referee two, makes excellent points that are very much appreciated! First of all, we agree that the paper is not 'strictly' a review, even though we cite a lot of papers. Given that insects are generally much better studied than spiders, we refer a lot to the insect literature and use it as a foundation on which to develop the possible similar connection with spiders. The problem, as the referee correctly stipulates, is that there are a lot of 'don't know's' or 'little data available' arguments that can be made when assessing demographic trends in spider populations, as well as the effects of various anthropogenic threats that are negatively affecting other species. I recently led a large review in Ecological Monographs in the Scientists' warning series of papers where me and approximately 70 international co-authors, addressed the effects of climate change and climatic extremes on insects. There is a lot more data there, making me somewhat mystified as to why such an important group of terrestrial organisms as spiders has been largely overlooked. This is what stimulated me and my PhD student Yuting Dong to initiate this paper, and, given a general paucity of studies available, we decided to make it both a review of existing literature on spider responses to heat and then to add our perspectives of areas relating to climatic extremes and spiders that we feel need urgent attention. These include sex-related responses to high temperatures, the role of spider ecology based on sex-related differences in body size (sexual size dimorphism) and other parameters like niche breadth, that have been little studied in spiders. Moreover, we argue that it is important to also fish out data sets collected over the years across the world to get some idea of demographic trends in spider abundance over time. Plenty of recent studies have reported sharp declines in insect taxa over the past several decades or more, but spiders for the most part have been ignored. The role of climate change and climatic extremes in driving temporal trends is, of course, very difficult to determine, but this is where controlled lab-based studies and even some field studies may help to evaluate spider responses to variable temperatures. I am currently involved in research addressing thermal responses in wolf spiders under variable temperature exposure. 

We have changed the title of the paper as also requested by referee two. We agree that the first one was both ambiguous and too directed, so the new one we think leaves the topic more 'open', adding the conjectural perspective suggested by this reviewer. 

We deeply apologize for the biological error in Figure 3! I take the biological realism of the insects and spiders I study VERY seriously, and I even wrote a paper about this several years ago, so when I make a similar error I humbly apologize for it. Indeed, I saw a paper in a taxonomic paper last year where the authors were describing a species of theridiid spider that was new to their region, and accompanied it with a photograph of a spider in an orb web! I was shocked that this had passed the peer-reviewers, so I am both happy and relieved that the referee correctly pointed out that orb-web spiders habitually sit head-downward in their webs and theridiids sit upside down. We have altered figure 3 to shown the orb-web weaver facing downwards. It was more difficult to do this for the tangle web weaving theridiid, so we hope it is acceptable to place the spider to also be facing head down in the illustration. Some of the Steatoda grossa and S. bipunctata spiders in my lab cultures do not sit completely horizontally upside down in their webs but are at a 45 degree angle and even occasionally are almost vertical, facing downwards. We hope this will suffice. 

The minor comments have all been incorporated into the revision (see yellow font).